# Role of Oxytocin/Vasopressin-Like Peptide and Its Receptor in Vitellogenesis of Mud Crab

**DOI:** 10.3390/ijms21072297

**Published:** 2020-03-26

**Authors:** Dongdong Lin, Yujie Wei, Haihui Ye

**Affiliations:** College of Ocean and Earth Sciences, Xiamen University, Xiamen 361102, China; 22320171150801@stu.xmu.edu.cn (D.L.); 22320181152145@stu.xmu.edu.cn (Y.W.)

**Keywords:** oxytocin/vasopressin, vitellogenesis, 17β-estradiol, mud crab

## Abstract

Oxytocin (OT)/vasopressin (VP) signaling system is important to the regulation of metabolism, osmoregulation, social behaviours, learning, and memory, while the regulatory mechanism on ovarian development is still unclear in invertebrates. In this study, *Spot*/*vp*-like and its receptor (*Spot*/*vpr-like*) were identified in the mud crab *Scylla paramamosain*. *Spot*/*vp*-like transcripts were mainly expressed in the nervous tissues, midgut, gill, hepatopancreas, and ovary, while *Spot*/*vpr*-like were widespread in various tissues including the hepatopancreas, ovary, and hemocytes. In situ hybridisation revealed that *Spot*/*vp*-like mRNA was mainly detected in 6–9th clusters in the cerebral ganglion, and oocytes and follicular cells in the ovary, while *Spot*/*vpr*-like was found to localise in F-cells in the hepatopancreas and oocytes in the ovary. In vitro experiment showed that the mRNA expression level of *Spvg* in the hepatopancreas, *Spvgr* in the ovary, and 17β-estradiol (E_2_) content in culture medium were significantly declined with the administration of synthetic *Sp*OT/VP-like peptide. Besides, after the injection of *Sp*OT/VP-like peptide, it led to the significantly reduced expression of *Spvg* in the hepatopancreas and subduced E_2_ content in the haemolymph in the crabs. In brief, *Sp*OT/VP signaling system might inhibit vitellogenesis through neuroendocrine and autocrine/paracrine modes, which may be realised by inhibiting the release of E_2_.

## 1. Introduction

Oxytocin (OT) and vasopressin (VP), the most ancient neuropeptides, were discovered from mammalian posterior pituitary in the early 1920s, having the oxytocic action or causing an increase in blood pressure [1]. Subsequently, they were purified and sequenced in 1953 before any other neuropeptides, and their preprohormones were firstly cloned from a rat genomic library in the early 1980s [2,3,4].

In mammals, the genes of OT and VP lie in the same chromosome towards each other, possessing a highly similar structure: a nonapeptide, located after a signal peptide and followed by a dibasic cleavage site and a neurophysin. The two nonapeptides both display a ring-like structure with an intramolecular disulfide bond (Cys1–Cys6), discriminated at position 8 where there is a neutral amino acid for OT, but a basic for VP [5,6]. And the neurophysin, with 14 cysteine residues, is an OT/VP-carrier protein which is essential to fold and sort the prohormones [7]. OT and VP are synthesised by the neurosecretory cells of supraoptic and paraventricular nucleus in hypothalamus and transported to the posterior pituitary via axons for secretion, which are multifunctional with essential roles in the regulation of social and reproductive behavior [8,9]. OT is mainly involved in inducing uterine contraction during parturition and milk ejection during lactation, and affecting parental care, social cognition, learning and memory [9,10,11], while VP is related to parental care, male/female pair bonding, blood pressure control and antidiuretic actions by stimulating water reabsorption via the renal collecting ducts [12]. The functions of OT/VP are mediated by the four receptors belonging to a family of G protein-coupled receptors (OTR, V_1a_R, V_1b_R and V_2_R) [13]. Among them, OTR, V_1a_R and V_1b_R all couple to the inositol trisphosphate (IP_3_)/Ca^2+^ signal transduction pathway, while V_2_R regulates the level of intracellular cAMP by the coupled adenylate cyclase [14,15,16,17]. Besides, OT/VP signaling system has also been reported in a number of other vertebrates such as birds, reptiles, amphibians and fishes, which all are supposed to have evolved from vasotocin, the ancestral nonapeptide, by gene duplication [18].

In invertebrates, OT or VP-like peptide has been widely identified among various phyla, while the co-occurrence of the two in the same species had never been demonstrated. In crustaceans, the OT/VP-like peptides have been reported in the blue swimming crab *Portunus pelagicus*, the water flea *Daphnia pulex*, the giant freshwater prawn *Macrobrachium rosenbergii* and the mud crab *Scylla paramamosain*, but lack of function tests [19,20,21,22]. Though the functional expression in cells and characterisation of OT/VP-like receptor have been investigated in mollusks, insects and nematodes, so far it isn’t studied in crustaceans [20,23,24,25].

In oviparous animals, the vitellogenesis, divided into autosynthesis and heterosynthesis according to the yolk origins, is an important process for the formation and accumulation of various yolk substances, which can provide nutrients and energy for the development of embryos and early larvae [26]. In crustaceans, the regulation on the ovarian maturation involves many neuropeptides. In the classic Panouse experiment in 1944, the removal of eyestalk in the shrimp *Palaemon serratus* resulted in enlarged ovaries and precocious eggs, suggesting that the effect was due to the removal of vitellogenesis-inhibiting hormone (VIH). Then it was isolated from the eyestalk of the American lobster *Homarus americanus* firstly and demonstrated to inhibit vitellogenesis [27,28]. Recently, crustacean female sex hormone (CFSH), isolated from the sinus gland of the blue crab *Callinectes sapidus*, is considered to be the female sex-determining hormone [29]. Besides in *S. paramamosain*, the red-pigment concentrating hormone (RPCH), secreted from the eyestalk ganglion, stimulates ovarian development, while the short neuropeptide F (sNPF) and allatostatin-C show inhibitory effect on ovarian development [30,31]. In *P. pelagicus*, it was reported that the *Pp*OT/VP-like peptide could inhibit the release of steroid hormones (17β-estradiol and progesterone) in ovarian explants in vitro [19]. Up to now, the molecular mechanism of OT/VP-like peptides on crustacean reproduction is unclear.

In this study, *Spot/vp* and *Spot/vpr*-like gene sequences were cloned from *S. paramamosain*, the tissue distribution and expression profiles during vitellogenesis were detected. Moreover, in vivo and in vitro experiments were performed to explore the role of the signaling system in vitellogenesis.

## 2. Results

### 2.1. Molecular Cloning and Bioinformatics of SpOT/VP-Like Preprohormone

The full-length sequence of *Spot*/*vp*-like (GenBank accession: MT123288) cDNA is 853 bp with 63 bp 5′ untranslated region (UTR), 313 bp 3′ UTR and 477 bp open reading frame (ORF). The preprohormone is composed of 158 amino acids containing a nonapeptide with the structure CFITNCPPGamide, which is located after a signal sequence and followed by a dibasic cleavage sites (GKR) and a neurophysin. The nonapeptide has a predicted molecular weight of 951.12 Da and a theoretical isoelectric point of 5.51, and the neurophysin has 14 cysteine residues (Figure A1A). Multiple sequence alignment suggests that the nonapeptides and three rigid cage-like domains are highly conserved: a neuropeptide-containing domain (one disulfide bonds) and two neurophysin domains (one four and the other three disulfide bonds) (Figure 1). Phylogenetic analysis show that *Sp*OT/VP-like preprohormone is clustered into the branch of crustaceans, then the branch with mollusks before insects (Figure A2).

### 2.2. Molecular Cloning and Bioinformatics of SpOT/VPR-Like Preprohormone

The *Spot*/*vpr*-like (GenBank accession: MT123289) cDNA ORF sequence is 1308 bp encoding 435 amino acids, which contains seven α-helical transmembrane (TM) regions with a predicted molecular weight of 48,067.19 Da and a theoretical isoelectric point of 7.01. The N-terminal domain is in the extracellular with two sites of N-linked glycosylation (Asn-*X*-Ser/Thr) and the C-terminal is in the intracellular. The preprohormone also contains some consensus sequences sites about phosphorylation for protein kinase C (Ser/Thr-*X*-Arg/Lys), casein kinase II (Ser/Thr-*X*-*X*-Asp/Glu) and cAMP-dependent protein kinase (Arg/Lys-Arg/Lys-*X*-Ser/Thr) (Figure A3) [32]. Multiple sequence alignment suggests that, except TM regions, the first and second extracellular loops, between TM_2_ and TM_3_ and between TM_4_ and TM_5_ respectively, have the highest homology, while the third intracellular loop shows hardly any sequence identity, in which *Sp*OT/VPR-like preprohormone contains ~20 amino acids more than vertebrate receptors (Figure 2).

### 2.3. Expression Profiles of Spot/vp and Spot/vpr-Like mRNA

RT-PCR shows that *Spot*/*vp*-like transcripts highly expressed in the nervous tissues, midgut, gill and epidermis, less express in the hepatopancreas and ovary, and hardly detected in the others. *Spot*/*vpr*-like are widespread in various tissues including the hepatopancreas, ovary and hemocytes (Figure 3A). qRT-PCR shows that there is a consistent temporal expression profile between *Spot*/*vp*-like transcripts in the nervous tissues and ovary and *Spot*/*vpr*-like in the ovary: highest expression level at pre-vitellogenic stage, significantly down-regulated at early and maintained lower at late (Figure 3B,C).

### 2.4. In Situ Hybridisation of Spot/vp and Spot/vpr-Like mRNA

The *Spot*/*vp*-like mRNA is mainly detected in 6th, 7th clusters of protocerebrum and 8th, 9th of deutocerebrum, and both in the oocytes and follicular cells of ovary (Figure 4A–I). The *Spot*/*vpr*-like is mainly localised in the F-cells of hepatopancreas and the oocytes in the ovary (Figure 4J–O). All control sections, probed with sense strand RNA, show no positive signal.

### 2.5. Synthetic SpOT/VP-Like Peptide Treatment In Vitro

In the hepatopancreas cultured in vitro, the mRNA expression level of *Spvg* is significantly down-regulated at 12 h with 10^−7^ M synthetic peptide administration (*p* < 0.05) (Figure 5A). The levels of E_2_ in the media is significantly decreased at 6 h, 10^−8^ M treatment and 12 h, 10^−7^ and 10^−8^ M treatments, respectively (*p* < 0.05), and extremely significantly declined at 12 h, 10^−6^ M treatment (*p* < 0.01) (Figure 5B). In the ovary, the mRNA expression level of *Spvgr* is extremely significantly down-regulated at 12 h, 10^−9^ M treatment (*p* < 0.01) (Figure 5C). The levels of E_2_ in the media is highly significantly lower at 6 h, 10^−9^ M treatment (*p* < 0.01), and significantly declined at 12 h, 10^−9^ M treatment, respectively (*p* < 0.05) (Figure 5D).

### 2.6. Synthetic SpOT/VP-Like Peptide Treatment In Vivo

The mRNA expression level of *Spvg* in the hepatopancreas and the E_2_ content in haemolymph are significantly down-regulated after the injection of synthetic peptide (*p* < 0.05) (Figure 6A,C). The mRNA expression levels of *Spvg* and *Spvgr* in the ovary were both not decreased significantly, but showed the inhibitory trends (Figure 6B). The oocyte diameter in the peptide injection group (52.8 ± 3.96 μm) was significantly shorter than that in crustacean physiological saline (CPS) group (74.6 ± 5.18 μm) (Figure 6D–G).

### 2.7. RNA Interference in the Hepatopancreas Explants In Vitro

In the hepatopancreas explants, the mRNA expression level of *Spot*/*vpr* was significantly down-regulated after the RNA interference (*p* < 0.05) (Figure 7A); simultaneously, the mRNA expression level of *Spvg* was significantly up-regulated with the treatment of synthetic peptide after RNA interference (*p* < 0.05) (Figure 7B).

## 3. Discussion

Hitherto, the study of OT/VP-like signaling system in crustacean reproduction has been rarely reported. In this study, it is suggested that the signaling system can inhibit vitellogenesis by means of neuroendocrine and autocrine/paracrine in *S. paramamosain*.

*Sp*OT/VP-like precursor sequence is more than 60% similar to those of other crustaceans such as *Portunus pelagicus* (GenBank accession: AUT12056.1) and *Penaeus vannamei* (GenBank accession: ROT67110.1). The molecular structure of the nonapeptides is highly conserved in various phyla, which makes it more universal to induce the functional response resulted from heterogenous OT/VPs. The bioactivity, resulted from *Conus* arg-conopressin-g/s in the brain of rat, is similar to those induced by vertebrate OT/VPs [33]. And the OT-like peptide from the pufferfish *Fugu rubripes* can functionally express in rat nerve cells [34]. Therefore, the discovery and functional screening of natural OT/VP peptides is an effective strategy for ligand selection and drug discovery, targeting at human OT/VP receptors [35]. OT/VP-like precursors in crustaceans clustered with those in mollusks firstly, then insects which belong to the same clade Pancrustacea. The strange result may be because that crustaceans are more closely related to the common ancestor of arthropods than insects in evolutionary status, such as *D. pulex*, considered to be the ancestor of insects with the divergence in ~420 million years ago [36]. Moreover, in order to adapt to a variety of environments during the evolution process from aquatic to terrestrial taxa, structures and functions of arthropods evolve over time. The OT/VP signaling system, lasting to crustaceans, is lost at least twice in insects which had been replaced by AKH, CCAP or Corazonin hormone systems, all participating in osmotic regulation and metabolism etc. [20].

The *Sp*OT/VPR-like is a G-protein coupled receptor with seven α-helical transmembrane regions. Except TMs regions, the first and second extracellular loops are the most conserved regions, connected by a disulfide bonds and important in ligand binding [37]. The sites of N-linked glycosylation in the extracellular N-terminal domain contribute to the target to plasma membrane, and the consensus sequences sites about phosphorylation are important in the modulation of G protein coupling and receptor function [32]. Compared with OT, *Sp*OT/VPR-like sequence is more similar to V_1a_R in mammals such as the mouse *Mus musculus* (GenBank accession: NP_058543.2) and the domestic horse *Equus caballus* (GenBank accession: XP_001917958.4) with homologies over 40%, suggesting that it may be a VP-like receptor. And when synthetic *Sp*OT/VP was added after knocking down *Spot*/*vpr* in vitro, the inhibitory effect of synthetic peptide was relieved, indicating that the receptor was *Sp*OT/VPR-like to some extent. It is the first time to clone and characterise OT/VPR-like in crustaceans. There are usually two sets of OT/VP signaling systems in molluscs, probably due to the additional gene replication [9,25]. Similarly, two OT/VP-like receptor (GenBank accession: ROT66533.1; XP_027226674.1) sequences, found in the genomic database of *P. vannamei*, suggested that multiple OT/VP signaling systems may also exist in crustaceans.

*Spot*/*vp*-like mRNA transcripts are mainly expressed in the nervous system, which is similar to the immunohistochemical results of invertebrates such as the hydrozoan *Hydra magnipapillata* and the polychaete *Neanthes japonica,* and the in situ hybridisation outcomes in the ant Camponotus fellah and the octopuses *Octopus vulgaris* [38,39,40,41,42]. The transcripts are also highly expressed in the midgut and gill, having the same expression pattern to *Ppot*/*vp*-like mRNA transcripts in *P. pelagicus* [19]. It is suggested that *Sp*OT/VP-like peptide can be involved in osmotic regulation, which is probably induced via crustacean hyperglycaemic hormone (CHH) as reported in the crab *Pachygrapsus marmoratus* and the crayfish *Astacus leptodactylus* [43,44]. *Spot*/*vpr*-like mRNA transcripts are expressed in various tissues, suggesting that *Sp*OT/VP signaling system is pleiotropic, similar to Lys-CP (from *Lymnaea stagnalis*) and OP (from *Octopus vulgaris*) [24,25,42].

In situ hybridisation shows that *Spot*/*vp*-like mRNA, similar to mollusk *op/ct*, is distributed in various kinds of neurons in cerebral ganglion, such as 6th and 7th neuronal clusters involved in reproductive regulation, and 8th and 9th in olfaction, learning and memory regulation [45,46,47]. *Spot*/*vpr*-like mRNA is expressed in F-cells of hepatopancreas, the main sites of enzyme synthesis, suggesting that the system may participate in the regulation of hepatopancreas on digestion, metabolism and reproduction. Besides, at early vitellogenic stage, *Spot*/*vp*-like mRNA distributes in oocytes/follicular cells, while *Spot*/*vpr*-like distributes in oocytes, indicating that *Sp*OT/VP-like from oocytes/follicular cells might act on oocytes. This result is different from the short neuropeptide F (sNPF) previously reported in the ovary in *S. paramamosain*, which showed that sNPF, synthesised by follicular cells, could act on oocytes/follicular cells [48]. Therefore, the communication might exist between oocytes and follicular cells in *S. paramamosain*, and we speculated that regulatory mode of autocrine/paracrine factors in the ovary is multifarious in crustaceans.

In mammals, both stimulatory and inhibitory effects of OT/VP/VTs on the release of E_2_/P_4_ (Progesterone), the vertebrate-type steroid hormones, from granulosa cells were reported [49,50,51]. Besides, in the catfish *Heteropneustes fossilis*, the vasotocin detected in granulosa cells of post-vitellogenic and absent in immature follicles, could promote the releases of E_2_, P_4_, P_4_ derivatives and prostaglandin, and cause the germinal vesicle breakdown and ovulation, which was suggested to correlate with the stage of luteinisation [52,53,54,55]. The E_2_ and P_4_-like were first discovered in crustaceans in 1978 [56]. Furthermore, the existence of E_2_/P_4_ in hepatopancreas, ovary and haemolymph, and the presence and activities of steroid-related enzymes (20α-HSD, 17β-HSD, 3β-HSD, 17β-HSD8) were recently reported in crustaceans, by which vitellogenesis was controlled [57,58,59,60,61,62]. Temporal expression profiles of *Spot*/*vp* and *Spot*/*vpr*-like mRNA showed that they might be involved in inhibiting ovarian development. Both in vitro cultured and in vivo long-term injection experiment showed that synthetic *Sp*OT/VP-like peptide could significantly inhibit the expression of *Spvg*/*Spvgr* in the hepatopancreas/ovary, respectively, which could be achieved by inhibiting the release of 17β-estradiol (E_2_) from the hepatopancreas/ovary. Histological analysis further showed that ovarian development was suppressed after the injection of synthetic peptide, with the significantly subduced oocyte growth. And it is probably resulted from less synthesis of yolk substances, and lack of accumulation of yolk substances in oocytes. Similarly, in *P. pelagicus*, the synthetic *Pp*OT/VP-like peptide also significantly inhibited the releases of E_2_ and P_4_ in the culture media from stage II and IV ovaries [19]. In conclusion, *Sp*OT/VP-like signaling system might inhibit vitellogenesis as neuroendocrine and autocrine/paracrine factors by inhibiting the release of E_2_ in *S. paramamosain*. This paper has enriched the researches on the OT/VP signaling system and the effect of neuropeptides on reproductive regulation in crustaceans. However, the detailed mechanisms on ovarian development should be clarified with more experiments in the future.

## 4. Materials and Methods

### 4.1. Animals

The female *S. paramamosain*, purchased at No.8 fishery market, Xiamen, China, were kept in plastic tanks containing sea water with aeration for 6 h at 27.0 ± 2.0 °C (temperature) and 27.0 ± 0.5 ppm (salinity), and fed with fresh clam *Ruditapes philippinarum*. In *S. paramamosain*, vitellogenesis can be classified into three stages according to the previous study: pre-vitellogenic, early vitellogenic and late vitellogenic stage [63]. All animal procedures were carried out in strict compliance with the National Institute of Health Guidelines for the Care and Use of Laboratory Animals. 

### 4.2. Molecular Cloning, Bioinformatics and Phylogenetics of SpOT/VP-Like Peptide and Its Receptor

Search for the nucleotide sequences of *Spot*/*vp*-like peptide and its receptor in the transcriptome of *S. paramamosain* cerebral ganglia, reported in previous studies, by BLAST against GenBank, non-redundant (Nr) and Gene Ontology (GO) [22]. The RNA, extracted from cerebral ganglia using Trizol reagent (Invitrogen, Carlsbad, CA, USA), was used for cDNA synthesis with the RevertAid^TM^ First Strand cDNA Synthesis Kit (Fermentas Inc., Ontario, Canada) and 5′/3′ RACE-cDNA synthesis with the SMART^TM^ RACE cDNA Amplification Kit (TaKaRa, Shiga, Japan). The full sequence was cloned according to the user manual for the SMART^TM^ RACE 5′/3′ Kit. All PCR products, separated by 1.0% (*w/v*) agarose gel electrophoresis and purified using agarose gel purification and extraction kit (Aidlab, Beijing, China), were cloned into the pMD19-T vector (Promega, Madison, WI, USA) and sequenced by Sangon Biotechnology Company Limited (Shanghai, China). ORFs and amino acid sequences were predicted by DNAStar software, and protein composition, molecular weight and isoelectric point were analysed by ExPASy software. The preprohormone sequences of OT/VPs and OT/VPRs from *S. paramamosain* against other species were obtained from NCBI, which were used to create the multiple sequence alignment by MEGA 7.0 software. Phylogenetic trees were carried out via the neighbor-joining method with 1000 bootstrap replicates using MEGA 7.0 software. Moreover, signal peptides were predicted by SignalP 5.0, cleavage sites, post-translational modifications and bioactive peptide products were predicted by NeuroPred, and transmembrane domains were predicted by TMHMM tool. All primers are listed in Table A1.

### 4.3. Tissue Distributions of Spot/vp and Spot/vpr-Like mRNA

The RNAs (*n* = 3), extracted from eyestalk ganglion, cerebral ganglion, thoracic ganglion, hepatopancreas, ovary, midgut, heart, stomach, gill, muscle, epicdermis and hemocytes at early vitellogenesis stage, were used for cDNA synthesis. Semi-quantitative RT-PCR was performed to detect the tissues distributions of *Spot*/*vp* and *Spot*/*vpr*-like transcripts. The reaction system was a volume of 25.0 μL containing 12.5 μL 2 × Premix Ex TaqΙΙ (Takara, Shiga, Japan), 1.0 μL cDNA, 0.5 μM each primer (10 μM) and 10.5 μL H_2_O. The thermocycling parameters were one cycle at 94 °C for 3 min, 35 cycles at 94 °C for 30 s, 57 °C for 30 s and 72 °C for 30 s, and a final extension at 72 °C for 10 min. The specificity control was performed using water as template and the internal control was performed using *β-actin*. All PCR products were separated by 1.5% (*w/v*) agarose gel electrophoresis with 1 × TAE Buffer (Solarbio, Beijing, China) and observed and photographed by gel imaging and analysis system (Bio-Rad, Shanghai, China). Three technical replicates were performed.

### 4.4. In Situ Hybridisation of Spot/vp and Spot/vpr-Like mRNA

The fragments of *Spot*/*vp* (310bp) and *Spot*/*vpr*-like (288bp) sequences were cloned into pGEM-T easy vector (Promega, Madison, WI, USA). The linear DNAs, amplified by PCR, were used as templates for riboprobes construction using DIG-Oligonucleotide Labeling Kit (Roche Molecular Biochemicals, Mannheim, Germany). Cerebral ganglia, hepatopancreas and ovaries at early vitellogenic stage were dissected and prepared for the paraffin-cut section. The serial seven-micron sections were used for Hematoxylin-eosin (H&E) staining and in situ hybridisation [64], visualised by the BCIP/NBT Chromogen Kit (Solarbio, Beijing, China) and mounted in Clear-Mount. All sections were observed and photographed by fluorescence confocal microscope (version, Axio Imager A2) equipped with digital camera (version, AxioCam MRc) (Carl Zeiss, Jena, Germany). Three technical replicates were performed.

### 4.5. Temporal Expression Profiles of Spot/vp and Spot/vpr-Like mRNA

The RNAs, extracted from nervous tissues and ovary at three vitellogenic stages (*n* = 5 per stage), were used for cDNA synthesis. Quantitative real-time PCR (qRT-PCR) was performed by a QuantStudio™ 6 Flex Real-Time PCR (Applied Biosystems) with SYBR^®^ Select Master Mix (TaKaRa, Shiga, Japan) according to the user manual. The internal control was performed using *β-actin*.

### 4.6. Synthetic SpOT/VP-Like Peptide Treatment In Vitro

The *Sp*OT/VP-like peptide (CFITNCPPG-NH2) used in this study was commercially synthesised by GL Biotechnology Company Limited (Shanghai, China). The crabs at early vitellogenic stage (*n* = 5) were anesthetised on ice for 10 min and sterilised in 75% ethanol for 10 min. Fragments of hepatopancreas and ovaries (~0.2 g) were dissected out, washed in CPS, and precultured at 26 °C in 24-wells plates with 200 μL Leibovitz’s L-15 medium (Gibco Invitrogen Corporation, Grand Island, NY, USA) each well. One hour later, substituted the medium with L-15 medium containing different concentration of synthetic peptide (0, 10^−9^, 10^−8^, 10^−7^ and 10^−6^ M) and incubated for 6 or 12 h. Then, collected the fragments for RNA extraction and cDNA synthesis. *β-actin* was used as the internal control, and qRT-PCR was performed to detect the relative transcript levels of *Spvg* and *Spvgr*. Meanwhile, culture media were collected and handled according to the steroid extraction protocol, and the level of E_2_ in the media were measured by DetectX^®^ SERUM 17β-ESTRADIOL Enzyme Immunoassay Kit (Arbor Assays, Ann Arbor, MI, USA). Three technical replicates were performed.

### 4.7. Synthetic SpOT/VP-Like Peptide Injection

The crabs at early vitellogenic stage were randomly divided into two groups (*n* = 5 per group), injected with 100 ng/g synthetic peptide or CPS, with the same volume, respectively. In addition, 5 crab without treatment were as initial control. Injections were given every four days for a total of 17 days duration, and on the 18th day, hepatopancreas and ovaries were collected and treated, and haemolymph was extracted and measured as above. Besides, the ovary tissues were dissected and prepared for the paraffin-cut section. The serial seven-micron sections were performed using H&E staining and the diameter of oocytes were measured. Three technical replicates were performed.

### 4.8. RNA Interference In Vitro

The fragments of *Spot*/*vpr*-like (478 bp) and *gfp* (454 bp) sequences were cloned into the pGEM-T easy vector. The linear DNAs, amplified by PCR, were used as templates for *dsRNA* transcribed by T7/SP6 polymerases. Hepatopancreas tissue was dissected out and treated as Section 2.7 with 200 μL L-15 media containing 5 μg *Spot*/*vpr dsRNA* or *gfp dsRNA*, and CPS with the same volume, respectively. After incubation at 26 °C for 10 h, substituted the media with 200 μL L-15 media containing 10^−8^ M synthetic peptide each well for 12 h. The hepatopancreas explants were collected for RNA extraction and cDNA synthesis. *β-actin* was used as internal control, and qRT-PCR was performed to detect the relative expression levels of *Spot*/*vpr* and *Spvg*. Three technical replicates were performed.

### 4.9. Statistical Analysis

The relative expression levels were calculated via the 2^−ΔΔCt^ algorithm. All data were expressed as mean ± SD and analysed using One-way ANOVA followed by Tukey’s multiple range tests, Dunnett’ s T tests and Student’s T-tests (SPSS Statistics 18.0) to estimate the statistical differences of mean levels among the groups with statistically significant at *p* < 0.05 and extremely significant at *p* < 0.01.

## Figures and Tables

**Figure 1 ijms-21-02297-f001:**
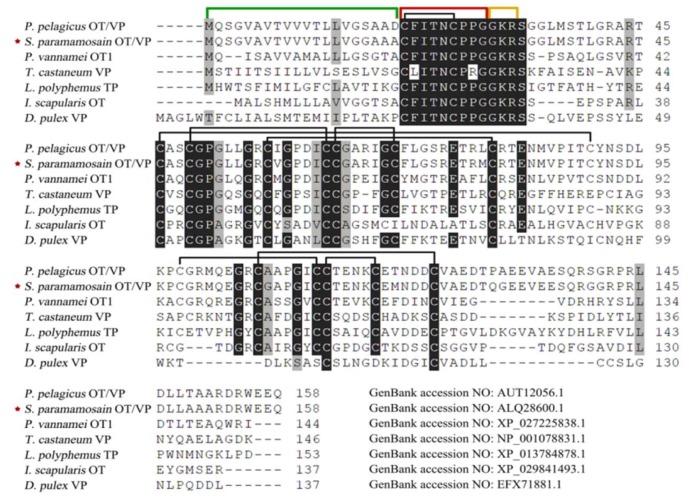
Multiple sequence alignment of deduced preprohormone sequences from *Sp*OT/VP-like peptide and other OT/VPs. Numbers of amino acid are listed on the right side. Nonapeptides, cleavage sites and signal peptides are marked in red, yellow and green semi-frames respectively and disulfide bonds are shown on black. OT: oxytocin; VP: vasopressin.

**Figure 2 ijms-21-02297-f002:**
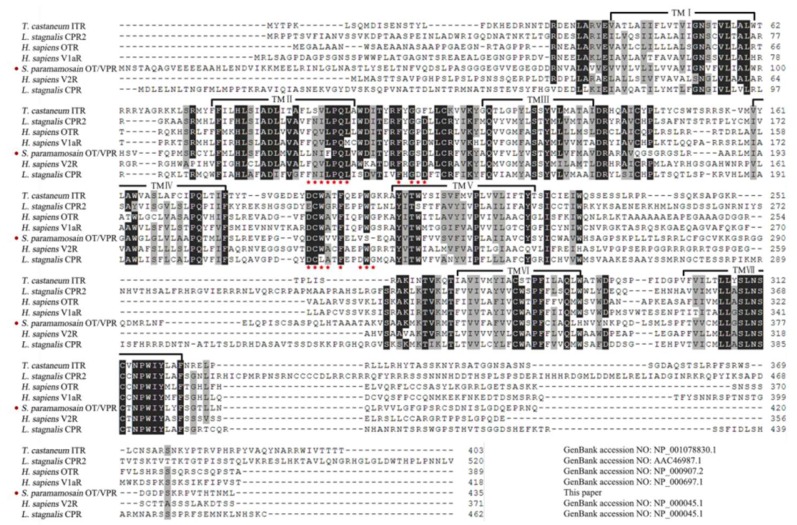
Multiple sequence alignment of deduced preprohormone sequences from *Sp*OT/VP-like receptor and other OT/VPRs. Numbers of amino acid are listed on the right side. Seven transmembrane domains are shown on black semi-frames.

**Figure 3 ijms-21-02297-f003:**
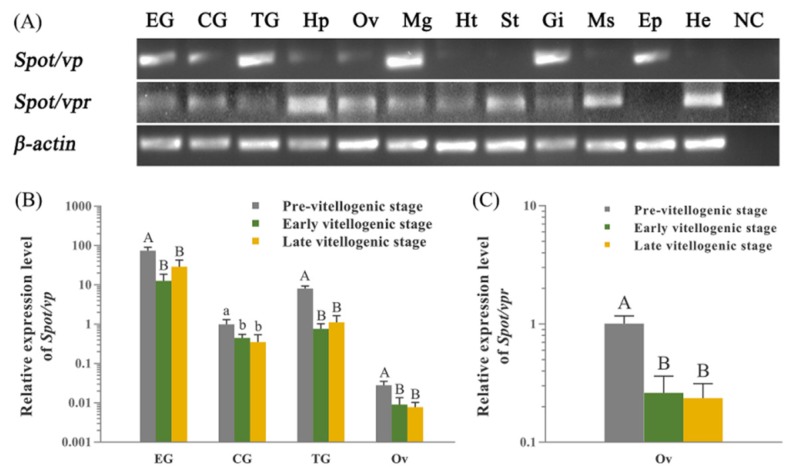
Expression analysis of *Spot*/*vp* and *Spot*/*vpr*-like transcripts in *S. paramamosain*. (**A**) Tissue distribution of *Spot*/*vp* and *Spot*/*vpr*-like transcripts in 12 tissues of early vitellogenic stage *S. paramamosain*: EG, eyestalk ganglion; CG, cerebral ganglion; TG, thoracic ganglion; Hp, hepatopancreas; Ov, ovary; Mg, midgut; Ht, heart; St, stomach; Gi, gill; Ms, muscle; Ep, epidermis; He, hemocyte; NC, negative control (a PCR reaction performed without adding template). (**B**) Expression analysis of *Spot*/*vp*-like transcripts at different vitellogenic stages in EG, CG, TG and Ov. (**C**) Expression analysis of *Spot*/*vpr*-like transcripts at different vitellogenic stages in Ov. *β-actin* as the reference gene. (“a and b”, *p* < 0.05; “A and B”, *p* < 0.01; one-way ANOVA followed by Tukey’ s multiple range tests; *n* = 5).

**Figure 4 ijms-21-02297-f004:**
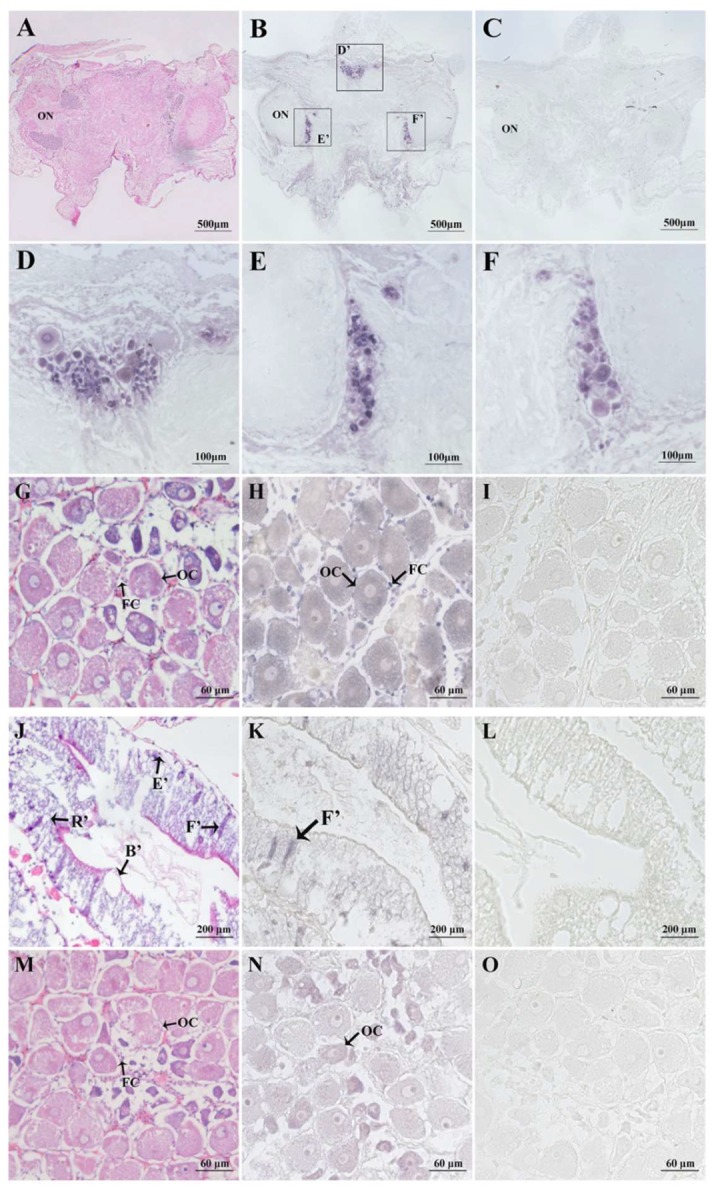
In situ hybridisation. (**A**–**F**) *Spot*/*vp*-like RNA of the cerebral ganglion. (**A**), Hematoxylin-eosin (H&E) staining; (**B**), antisense probe; (**C**), sense probe; **D**), the cell cluster 6, 7; (**E**,**F**), the cell cluster 8, 9. ON, olfactory neuropile. (**G**–**I**) *Spot*/*vp*-like RNA of the ovary. (**G**), Hematoxylin-eosin (H&E) staining; (**H**), antisense probe; (**I**), sense probe. OC, oocytes; FC, follicular cells. (**J**–**L**) *Spot*/*vpr*-like RNA of the hepatopancreas. (**J**), Hematoxylin-eosin (H&E) staining; (**K**), antisense probe; (**L**), sense probe. B’, B-cell; F’, F-cell; R’, R-cell; E’, E-cell. (**M**–**O**): *Spot*/*vpr*-like RNA of the ovary. (**M**), Hematoxylin-eosin (H&E) staining; (**N**), antisense probe; (**O**), sense probe.

**Figure 5 ijms-21-02297-f005:**
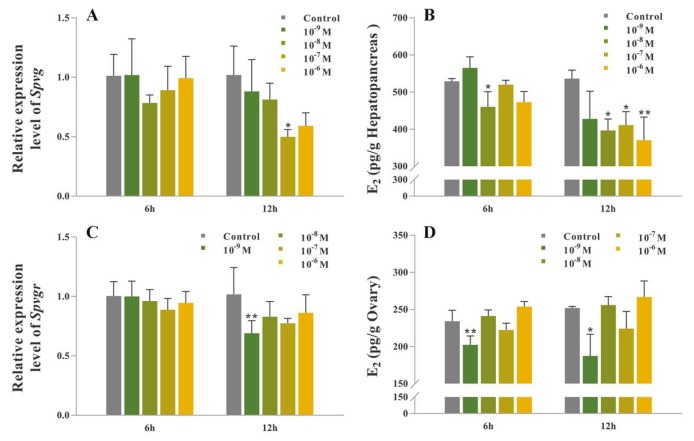
Effects of synthetic *Sp*OT/VP-like peptide on gene expressions and E_2_ contents in vitro. (**A**) Relative expression level of *Spvg* in the hepatopancreas. (**B**) Release of E_2_ from the hepatopancreas. (**C**) Relative expression level of *Spvgr* in the ovary. (**D**) Release of E_2_ from the ovary. *β-actin* as the reference gene. (* *p* < 0.05; ** *p* < 0.01; one-way ANOVA followed by Dunnett’ s T tests; n = 5).

**Figure 6 ijms-21-02297-f006:**
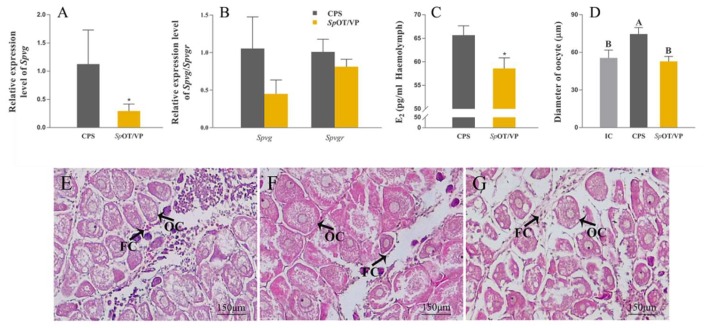
Effects of synthetic *Sp*OT/VP-like peptide on gene expressions, E_2_ contents and ovarian development in vivo. (**A**) The relative expression level of *Spvg* in the hepatopancreas. (**B**) The relative expression level of *Spvg* and *Spvgr* in the ovary. (**C**) The content of E_2_ in the haemolymph. *β-actin* as the reference gene. (* *p* < 0.05; ** *p* < 0.01; one-way ANOVA followed by Dunnett’ s T tests; n = 5). (**D–G**) Histological changes of ovarian development. (**D**), diameter of oocytes; (**E**), initial control (IC); (**F**), negative control (CPS); (**G**), group injected with synthetic peptide. OC, oocytes; FC, follicular cells. (“a and b”, *p* < 0.05; “A and B”, *p* < 0.01; one-way ANOVA followed by Tukey’ s multiple range tests; *n* = 5).

**Figure 7 ijms-21-02297-f007:**
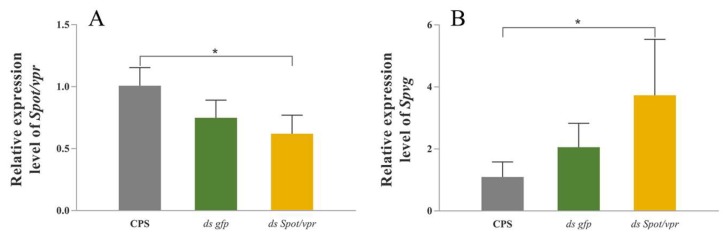
Effects of RNAi in the hepatopancreas explants in vitro. (**A**) Effects of RNAi with the application of *Spot*/*vpr dsRNA*. (**B**) Relative expression level of *Spvg* with the application of synthetic *Sp*OT/VP-like peptide after RNAi. *β-actin* as the reference gene. (* *p* < 0.05; one-way ANOVA followed by Tukey’ s T tests; n = 5).

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
