# Peer review of "Role of Oxytocin/Vasopressin-Like Peptide and Its Receptor in Vitellogenesis of Mud Crab"

_ijms, 2020, doi:10.3390/ijms21072297_

Round 1

Reviewer 1 Report

This report presents the characterization of a peptide of the neurohypophysial hormone family in the mud crab. Although the family is conserved in invertebrates as well as vertebrates, the understanding of functions of the crustacean ortholog is still lacking. Therefore, results of this study will give us useful information how this peptide hormone system regulates crustacean physiology.

Followings are my comments and questions for the authors.

1) The authors should give the rationale for analyzing E2. The relationship between E2/ERs and OT/OTR is well understood in mammals and fish. And actually, functions of E2 (and P4) have been studied on some crab and shrimp species in terms of the regulation of vitellogenesis. However, the vertebrate-type steroid hormones seem to be controversial in Crustaceans. Do the recent sequencing data in public database and related reports of Scylla species (e.g. https://doi.org/10.1016/j.ygcen.2016.03.008 and https://doi.org/10.1016/j.aquaculture.2019.734447) suggest the actual existence of such sex steroid molecule(s) and the steroidogenesis pathways in decapod crustaceans?

And what is the reason for the difference in time-course change in E2 production/secretion from the hepatopancreas and ovary?

2) In discussion on the expression of Spot/vp-like in central nervous system, the authors are citing old immunohistochemical data (location of peptides in secretory granule) and recent data but for the other peptide hormone. In addition to them, some reports using in situ hybridization (location of neuronal cell body) on related peptides and species (e.g. inotocin of arthropods; https://doi.org/10.1073/pnas.1817788116) should be cited and discussed.

3) Regarding the RNAi experiment using the hepatopancreas explants (Fig. 7), the difference among all groups should be analyzed using multiple comparison test. The results of experiments using RNAi is somehow and sometimes tricky. If there were no statistical difference between dsGFP group and dsSpot/vpr group, the difference in Spot/vpr expression and SpVg would not be attributed to the addition of dsSpot/vpr. Is 10-hr dsRNA treatment enough to achieve the clearance of existed receptor protein?

Specific comments:

L130 and L173: “a and b” and “A and B”.

L220: The two precursors have the same sequence in mature hormone and following neurophysin. As far as I know, only a single molecular species of OT/VP-like peptide is reported in various transcriptome analyses on Arthropods. So, it is better to describe only the two receptors to suggest multiple OT/VP signaling systems here.

Fig. 6E-G: Pictures with higher magnification are better to show follicular cells.

Figures: Gene names in Y-axis title should be Spot/vpr or Spot/vpr-like like in the main text. And in the same rule, SpVg and SpVgr will be Spvg and Spvgr.

Author Response

Response to Reviewer 1 Comments

We are pleased to note the favorable comments from referees. Those comments are all valuable and very helpful for revising and improving our paper. Revised portion are marked in red in the paper. We have made the response to the comments as below:

Point 1: The authors should give the rationale for analyzing E2. The relationship between E2/ERs and OT/OTR is well understood in mammals and fish. And actually, functions of E2 (and P4) have been studied on some crab and shrimp species in terms of the regulation of vitellogenesis. However, the vertebrate-type steroid hormones seem to be controversial in Crustaceans. Do the recent sequencing data in public database and related reports of Scylla species (e.g. https://doi.org/10.1016/j.ygcen.2016.03.008 and https://doi.org/10.1016/j.aquaculture.2019.734447) suggest the actual existence of such sex steroid molecule(s) and the steroidogenesis pathways in decapod crustaceans?

And what is the reason for the difference in time-course change in E2 production /secretion from the hepatopancreas and ovary?

Response 1: Thank you for your comments. The rationale for analyzing E2 has been added as follows. “In mammals, both stimulatory and inhibitory effects of OT/VP/VTs on the release of E2/P4 (Progesterone), the vertebrate-type steroid hormones, from granulosa cells were reported [49-51]. Besides, in the catfish Heteropneustes fossilis, the vasotocin detected in granulosa cells of post-vitellogenic and absent in immature follicles, could promote the releases of E2, P4, P4 derivatives and prostaglandin, and cause the germinal vesicle breakdown and ovulation, which was suggested to correlate with the stage of luteinization [52-55]. The E2 and P4-like were first discovered in crustaceans in 1978 [56]. Furthermore, the existence of E2/P4 in hepatopancreas, ovary and haemolymph, and the presence and activities of steroid-related enzymes (20α-HSD, 17β-HSD, 3β-HSD, 17β-HSD8) were recently reported in crustaceans, by which vitellogenesis was controlled [57-62].”. Related studies have been cited (L248-L257).

In crustaceans, hepatopancreas is a multifunctional tissue playing important roles in digestion, absorption, metabolism, immunity and reproduction (Rőszer, 2014), while ovary is mainly related to reproduction. And E2 can participates in multiple functions such as reproduction, lipid metabolism, etc. (Liu, et al., 2019), of which the role is more complex in hepatopancreas. In addition, the vitellogenesis is divided into autosynthesis and heterosynthesis in Scylla paramamosain. The heterosynthesis in hepatopancreas is later and more important than the autosynthesis in ovary. The demand and sensitivity to E2 between hepatopancreas and ovary are probably different. Therefore, we think it is understandable that the time-course of E2 production/secretion varies between hepatopancreas and ovary.

Reference:

Rőszer, T. The invertebrate midintestinal gland (“hepatopancreas”) is an evolutionary forerunner in the integration of immunity and metabolism. Cell and tissue research 2014, 358, 685-695.

Liu, M. M.; et al. Effect of estradiol on hepatopancreatic lipid metabolism in the swimming crab, Portunus trituberculatusGen. Comparative. Endocrinol. 2019, 280, 115-122.

Point 2: In discussion on the expression of Spot/vp-like in central nervous system, the authors are citing old immunohistochemical data (location of peptides in secretory granule) and recent data but for the other peptide hormone. In addition to them, some reports using in situ hybridization (location of neuronal cell body) on related peptides and species (e.g. inotocin of arthropods; https://doi.org/10.1073/pnas.1817788116) should be cited and discussed.

Response 2: Thanks for your comments and we agree with you to have any discussion from multiple levels. The reports on related peptides and species have been cited and discussed as suggested (L227-L228).

Reference: 

Akiko, K.; Naoto, M.; Hiroki, T.; et al. Oxytocin/vasopressin-like peptide inotocin regulates cuticular hydrocarbon synthesis and water balancing in ants. Proc. Natl. Acad. Sci. USA. 2019, 116,  5597-5606.

Takuwa-Kuroda, K.; Iwakoshi-Ukena, E.; Kanda, A.; et al. Octopus, which owns the most advanced brain in invertebrates, has two members of vasopressin/oxytocin superfamily as in vertebrates. Regul. Pept. 2003115, 139–149

Point 3: Regarding the RNAi experiment using the hepatopancreas explants (Fig. 7), the difference among all groups should be analyzed using multiple comparison test. The results of experiments using RNAi is somehow and sometimes tricky. If there were no statistical difference between dsGFP group and dsSpot/vpr group, the difference in Spot/vpr expression and SpVg would not be attributed to the addition of dsSpot/vpr. Is 10-hr dsRNA treatment enough to achieve the clearance of existed receptor protein?

Response 3: Thank you for your comments. The difference among all groups has been analyzed using Tukey's multiple range tests as suggested (L182). Unfortunately, there was no statistical difference between dsgfp and dsSpot/vpr groups. It is indeed somehow and sometimes tricky and we are still exploring the reasons. However, no difference between CPS and dsgfp groups, but a statistical difference between CPS and dsSpot/vpr groups might show that the difference was caused by the addition of dsSpot/vpr to some extent. In fact, RNAi had been tried at 4/8/10/12/16 h in this study, among which 10 h treatment showed the best silence efficiency. Time setting referred to the previous researches (Gong, et al., 2015; Yang, et al., 2014) and the increase of Spvg mRNA expression level in dsSpot/vpr group, compared with CPS, was also displayed that existed receptor protein had been almost cleared.

Reference: 

Gong, J.; et al. Ecdysteroid receptor in Scylla paramamosain: a possible role in promoting ovary development. Journal of Endocrinology 2015, 224, 273-287.

Yang, Y.; et al. Immune responses of prophenoloxidase in the mud crab scylla paramamosain against vibrio alginolyticus infection: in vivo and in vitro gene silencing evidence. Fish & Shellfish Immunology 2014, 39, 237-244.

Specific comments:

Point 4: L130 and L173: “a and b” and “A and B”.

Point 5: L220: The two precursors have the same sequence in mature hormone and following neurophysin. As far as I know, only a single molecular species of OT/VP-like peptide is reported in various transcriptome analyses on Arthropods. So, it is better to describe only the two receptors to suggest multiple OT/VP signaling systems here.

Point 6: Fig. 6E-G: Pictures with higher magnification are better to show follicular cells.

Point 7: Figures: Gene names in Y-axis title should be Spot/vpr or Spot/vpr-like like in the main text. And in the same rule, SpVg and SpVgr will be Spvg and Spvgr.

Response 4-7: Thank you for your comments and suggestions. 4) and 5): Corrected accordingly (L130 and L174; L221); 6): Fig. 6E-G with higher magnification has been re-uploaded (L167); 7): Gene names (Spot/vp-like, Spot/vpr-like, Spvg and Spvgr) have been modified to be internally consistent within the manuscript. 

Reviewer 2 Report

The present paper analyzes the role of oxytocin in the modulation of vitellogenesis of the crab Scylla paramamosai.
Authors, in section 4.2, mentioned the building of a local transcriptome of cerebral ganglia in S. paramamosain. However, they did not mention any information on the quality check and the parameters applied for selecting the raw reads, the software used for the assembly, the approaches used for the final quality assessment of the transcriptome (duplicates presences, average length, etc.) and how they selected the differentially expressed genes. If a published paper on this is already availabe, authors could add the reference, but if the transcriptomic work has not yet been published, they need to add further information on the approach applied for genereting the transcriptome. English should be carefully reviewed throughout the text.
In section 4.3 is lacking the taq polymerase used, the PCR reagents’ final concentrations and the final volume of each reaction. Also, authors did not mention the number of technical replicates used for each biological sample.
Authors claimed they obtained OT/VPs and OT/VPRs sequences from NCBI (GenBank, I guess). They should also provide the GenBank reference of them.
As far the statistical analysis is concerned (section 4.9), when the authors compare more than 2 groups, they should check data for normality and homogeneity of variance before using ANOVA and post-hoc Tukey test if they decide to use parametric statistics, or Kruskal-Wallis rank sum test followed by post-hoc Wilcoxon rank sum test pairwise comparisons if they decide for non-parametric one.
English and style should be reviewed.

Some minor comments:
Line 119: nervous tissues not organ.
Line 122: Y axis title: "Relative expression level" in all figures.
Line 137: figure is of good quality, please increase the size.
Line 152: "is highly significantly lower" instead of "is extremely significantly decreased".
Line 159: why not "treatment in vivo" rather than "injection"?
Line 190: it is not common to begin a sentence with and.
Line 199: ... belong to the same clade Pancrustacea.
Line 203: hydrophytic? The authors mean from aquatic to terrestrial taxa?
Lines 204-205: ... crustaceans which almost belong to terraneous taxa? This sentence is wrong. The paragraph from line 199 to line 206 should be rewritten.
Line 217: It is the first to clone... what? The sentence is wrong.
Line 227: same as line 190.
Line 223: nervous system not "organ".
Line 276: was instead of were.
Line 284: please add the reference for Protparam software.
Line 300: authors did not mention the running buffer used for the gel electrophoresis.
Line 311: please mention the microscope and camera names.
Line 334: 3 groups on 5 individuals: 1 injected with synthetic peptide, 1 sham injected, the third group?
Lines 380-382: I can't find any reference on the method and software used.

Author Response

Response to Reviewer 2 Comments

We are pleased to note the favorable comments from referees. Those comments are all valuable and very helpful for revising and improving our paper. Revised portion are marked in red in the paper. We have made the response to the comments as below:

Point 1: Authors, in section 4.2, mentioned the building of a local transcriptome of cerebral ganglia in S. paramamosain. However, they did not mention any information on the quality check and the parameters applied for selecting the raw reads, the software used for the assembly, the approaches used for the final quality assessment of the transcriptome (duplicates presences, average length, etc.) and how they selected the differentially expressed genes. If a published paper on this is already availabe, authors could add the reference, but if the transcriptomic work has not yet been published, they need to add further information on the approach applied for genereting the transcriptome. English should be carefully reviewed throughout the text.

Response 1: Thank you for pointing out the lack of information. The paper on the transcriptome building used in this study, is already published and it has been cited in section 4.2 (L288-L290).

Thanks for your suggestions. The manuscript has been carefully reviewed and corrected.  

Point 2: In section 4.3 is lacking the taq polymerase used, the PCR reagents’ final concentrations and the final volume of each reaction. Also, authors did not mention the number of technical replicates used for each biological sample.

Authors claimed they obtained OT/VPs and OT/VPRs sequences from NCBI (GenBank, I guess). They should also provide the GenBank reference of them.

Response 2: I am sorry for the carelessness. The detailed information of PCR has been supplemented in section 4.3 (L311-L313). The number of technical replicates has been added for each biological sample (L318, L330, L351, L360, L371).

Spot/vp and Spot/vpr-like sequences, cloned by ourselves, had been uploaded to NCBI (GenBank accession: MT123288, MT123289) (L81, L102).

Point 3: As far the statistical analysis is concerned (section 4.9), when the authors compare more than 2 groups, they should check data for normality and homogeneity of variance before using ANOVA and post-hoc Tukey test if they decide to use parametric statistics, or Kruskal-Wallis rank sum test followed by post-hoc Wilcoxon rank sum test pairwise comparisons if they decide for non-parametric one.

English and style should be reviewed.

Response 3: Thank you for your comments. All data had been checked for normality and homogeneity of variance before statistical analyses. They have been analyzed using ANOVA followed by Tukey's multiple range tests as suggested (SPSS 18.0) (L376, L122, L167, L182).

Some minor comments:

Point 4: Line 119: nervous tissues not organ.

Point 5: Line 122: Y axis title: "Relative expression level" in all figures.

Point 6: Line 137: figure is of good quality, please increase the size.

Point 7: Line 152: "is highly significantly lower" instead of "is extremely significantly decreased".

Point 8: Line 159: why not "treatment in vivo" rather than "injection"?

Point 9: Line 190: it is not common to begin a sentence with and.

Point 10: Line 199: ... belong to the same clade Pancrustacea.

Response 4-10: Thanks for your carefully checking for these grammatical errors and advice for figures. All of them have been rectified as suggested. 4): L119; 5): Corrected accordingly in all figures (L122, L154, L167, L182); 6): L137; 7): L151; 8): L160; 9): L193; 10): L200.

Point 11: Line 203: hydrophytic? The authors mean from aquatic to terrestrial taxa?

Point 12: Lines 204-205: ... crustaceans which almost belong to terraneous taxa? This sentence is wrong. The paragraph from line 199 to line 206 should be rewritten.

Response 11-12: I am very sorry for the misuse of words. 11): The sentence has been revised as: “... from aquatic to terrestrial taxa” (L204-L205); 12): “... belong to terraneous taxa” should be “... belong to aquatic taxa”, and the paragraph has been rewritten as: “ Moreover, in order to adapt to a variety of environments during the evolution process from aquatic to terrestrial taxa, structures and functions of arthropods evolve over time. The OT/VP signaling system, lasting to crustaceans, was lost at least twice in insects which had been replaced by AKH, CCAP or Corazonin hormone systems, all participating in osmotic regulation and metabolism etc [20].” (L201-L208).

Point 13: Line 217: It is the first to clone... what? The sentence is wrong.

Response 13: Thank you for your correction. The sentence has been modified as: “It is the first time to clone and characterize OT/VPR-like in crustaceans” (L219).

Point 14: Line 227: same as line 190.

Point 15: Line 223: nervous system not "organ".

Point 16: Line 276: was instead of were. 

Point 17: Line 284: please add the reference for Protparam software.

Response 14-17: Rectified as suggested. 14): L231; 15): L225; 16): L291; 17): L299.

Point 18: Line 300: authors did not mention the running buffer used for the gel electrophoresis.

Point 19: Line 311: please mention the microscope and camera names.

Response 18-19: I am sorry for the lack of information. They have been added as suggested. 18): L317; 19): L329-L330.

Point 20: Line 334: 3 groups on 5 individuals: 1 injected with synthetic peptide, 1 sham injected, the third group?

Response 20: Thank you for pointing out the unclear expression. The third group is an initial control without treatment. The sentence has been revised (L354-L356).

Point 21: Lines 380-382: I can't find any reference on the method and software used.

Response 21: I am sorry for our negligence. “Phylogenetic trees were carried out via the neighbor-joining method with 1000 bootstrap replicates using MEGA 7.0 software.” has been supplied (L404-L406).